# Serological Testing for Mycobacterial Heat Shock Protein *Hsp65* Antibody in Health and Diseases

**DOI:** 10.3390/microorganisms8010047

**Published:** 2019-12-25

**Authors:** Peilin Zhang, Lawrence M. Minardi, John Todd Kuenstner, Sylvia T. Zhang, Steve M. Zekan, Rusty Kruzelock

**Affiliations:** 1PZM Diagnostics, LLC, Charleston, WV 25301, USA; eyemanmd@hotmail.com (L.M.M.); talewkuenstner@aol.com (J.T.K.); smzekan@gmail.com (S.M.Z.); 2Helen Diller Family Comprehensive Cancer Center, University of California, San Francisco Medical Center at Mission Bay, San Francisco, CA 94158, USA; sylviatinazhang@gmail.com; 3West Virginia Regional Technology Park, Union Carbide Road, South Charleston, WV 25309, USA; rusty.kruzelock@wvrtp.com

**Keywords:** *Mycobacterium*, heat shock protein 65, Crohn’s disease, Sjogren’s syndrome, autoimmune diseases

## Abstract

Mycobacterial heat shock protein 65 gene (*Hsp65*) has been widely used for classification of Mycobacterial species, and detection of Mycobacterial genes by molecular methods and has proven useful in identification of Mycobacterial infection in various clinical conditions. Circulating antibody against Mycobacterial *hsp65* has been found in many clinical diseases including autoimmune diseases (Crohn’s disease, lupus erythematosus, multiple sclerosis, diabetes, etc.), atherosclerosis and cancers. The prevalence of anti-*Hsp65* antibody in the normal healthy population is unknown. We determined the blood levels of antibody against Mycobacterial *hsp65* in the normal population represented by 288 blood donors of the American Red Cross and tested the blood of 109 patients with Crohn’s disease and 28 patients with Sjogren’s syndrome for comparison. The seroprevalence of anti-*Hsp65* IgG in the normal population of Red Cross donors was 2.8% (8 of 288 positive). The *Hsp65* antibody levels were significantly elevated in patients with Crohn’s disease and Sjogren’s syndrome. The prevalence of *Hsp65* antibody in Crohn’s disease patients was 67.9% (74 of 109 patients), and 85.7% for Sjogren’s patients (24 of 28 patients). Our data indicate that anti-*Hsp65* antibody is rare in the normal population, but frequent in chronic diseases. The presence of circulating *Hsp65* antibody reflects an abnormal immune (adaptive) response to Mycobacterial exposure in patients with chronic diseases, thus differentiating the patients with chronic diseases from those clinical mimics.

## 1. Introduction

Heat shock proteins (*Hsps*) are a family of chaperonins in all living organisms playing critical roles in protein folding [1,2]. The heat shock proteins are responsive to stress related conditions and constitute a stress-responsive molecular pathway in all living cells [1,2]. *Hsp65* of *Mycobacterium* is known to be present in all mycobacterial species, including *Mycobacterium avium* ssp. *paratuberculosis* (MAP) and *Mycobacterium avium* ssp. *hominissuis* (MAH), and the presence of the *Hsp65* gene has been used as a molecular marker for identification of mycobacterial infection [3,4]. Sequencing of *Hsp65* genes within a family of mycobacteria can be used for sub-classification of mycobacterial species [3,4,5]. In bacterial species, *Hsp65* is known as GroEL, and the human homologue is known as human Hsp60, with similar molecular functions in both bacterial and human cells [6,7]. Previous studies demonstrated the presence of anti-*Hsp65* antibodies within the circulation of patients with a variety of chronic conditions, including autoimmune diseases such as multiple sclerosis (MS), systemic lupus erythematosus (SLE), and Crohn’s disease (CD), as well as other chronic conditions such as heart disease, atherosclerosis, and different types of cancers [8,9,10,11]. In addition, meta-analyses have concluded that the prevalence of MAP infection is significantly higher in patients with CD as compared to the normal healthy control population [12,13].

Anti-*hsp65* antibodies of likely mycobacterial antigenic origin are closely related to human anti-Hsp60 antibodies, and anti-Hsp60 antibodies of human antigens are considered “autoantibody”; thus, anti-Hsp60 antibody and the anti-*Hsp65* antibody within the circulation of patients are likely identical, responding to the same or similar antigens through molecular mimicry [8,14,15,16]. Whether the antigens of these two antibodies are identical remains largely controversial, as mycobacterial antigens are likely infectious from the environment, and the human antigens are likely autoimmune with unclear mechanisms [16]. The advance of human microbiome studies showed the presence of numerous commensal bacteria within the body, and these commensal bacteria also express the heat shock stress response pathway, including *Hsp65* and other members of the same family [16]. However, the question of *Hsp65* antigenic origins remains unanswered. All the human chronic conditions occur in genetically susceptible individuals with environmental triggers. Is the antigenic origin a bacterium in the host or a *Mycobacterium* from the environment or the human cell itself?

In order to address these questions, we surveyed the normal population for seroprevalence of *Hsp65* antibody in blood donors of the American Red Cross, and compared the seroprevalence with those of chronic conditions, such as Crohn’s disease and Sjogren’s syndrome [17]. Our findings are supportive of the view that chronic diseases are infectious diseases in genetically susceptible individuals and the *Hsp65* antibodies in circulation represents an abnormal adaptive immune response to antigens of mycobacterial (environmental) origin. 

## 2. Materials and Methods

### 2.1. Direct ELISA Assays

#### 2.1.1. A: Recombinant Protein Biosynthesis

Recombinant Mycobacterial *Hsp65* protein was synthesized by Genscript Corporation (http://www.genscript.com/) through contract work commercially. The *Hsp65* protein from *Mycobacterial avium* ssp. *hominissuis* (MAH) was identified at PZM Diagnostics through a series of identification and mass spectrometry as described previously [14,18]. The genes encoding for *Hsp65* proteins of mycobacterial species (MAH) were synthesized chemically at Genscript Corp., and cloned into pUC57 bacterial expression vectors with His-tag using proprietary technology at Genscript Corp. The recombinant *Hsp65* protein was expressed in the *Escherichia coli* expression system and purified through His-tag columns. The final *Hsp65* protein was analyzed by Western blot analysis using anti-His-tag antibody and Coomassie blue stain of SDS-polyacrylamide gel for quality assurance. The purified recombinant *Hsp65* protein was delivered on dry ice and the protein concentration was adjusted at 10 μM with phosphate buffered saline (PBS) with 20% glycerol for storage.

#### 2.1.2. B: Establishment of Direct ELISA Assay

Our direct ELISA assays for human antibody testing is based on the Biolegend protocol with modification (https://www.biolegend.com/protocols/sandwich-elisa-protocol/4268/). Direct ELISA assays for human antibody levels against various recombinant microbial antigens were established based on a series of antigen titration and stringent detergent levels. The optimal concentration of the recombinant antigens for coating the 96-well plates was determined to be in the range of 0.1 to 1.0 nmol (1–10 ng/mL), and the plate coating was optimal at 4 °C over night. The carbonate coating buffer, blocking agents and concentration, washing buffer, substrate (TMB), and stop buffer were all based on the protocol from BioLegend (San Diego, CA, USA) (https://www.biolegend.com/protocols/sandwich-elisa-protocol/4268/). The data was collected by using the VersaMax ELISA plate analyzer from Molecular Device at OD450 TMB using the end-point protocol. The data was analyzed and plotted by using various components of R-statistics (Rstudio) package (http://statistics4everyone.blogspot.com/).

### 2.2. Blood Donor Units and Sample Processing

The study is exempt from Institutional Review Board (IRB) approval according to section 46.101(b) of 45CFR 46 which states that research involving the study of existing pathological and diagnostic specimens in such a manner that subjects cannot be identified is exempt from the Department of Health and Human Services Protection of Human Research Subjects. This determination was made by the IRB in WV, reviewed under protocol number #1194084 (2017). Red Blood Cells (RBCs) from the American Red Cross consist of erythrocytes concentrated from whole blood donation or collected by apheresis [19]. Based on the guideline from the American Red Cross, the RBC units contain citrate anticoagulant and usually one of several types of preservative solutions. The blood donors of the Red Cross have been screened under strict regulations in the US and the segments of the donor units are the identical blood units that have been transfused to the patients. The segments are usually retained at the hospital blood banks for 3 months at 4 °C before being discarded after transfusion. Depending on the preservative-anticoagulant, the hematocrit (Hct) of RBCs is about 55–65% for additive solutions (AS), AS-1, AS-3, AS-5, and AS-7, and about 65–80% for citrate–phosphate–dextrose adenine solutions, CPDA-1, CPD, and CP2D. RBCs contain 20–100 mL of donor plasma, usually less than 50 mL, in addition to preservative and anticoagulant. The typical volume of AS RBCs including additive solution is 300–400 mL. Each unit contains approximately 50–80 g of hemoglobin (Hgb) or 160–275 mL of red cells, depending on the Hgb level of the donor, the whole blood collection volume, and the collection and processing methods [19]. Leukocyte-reduced RBCs must retain at least 85% of the original RBCs. Each unit of RBCs contains approximately 250 mg of iron, almost entirely in the form of Hgb and the iron level varies depending on the original volume and concentration of the unit. There are a few tubal segments attached to the units containing the same RBCs, and these tubal segments are to be used for quality assurance testing or additional studies once the blood unit is transfused. The segments of the donor blood units were obtained from the local Red Cross offices after the blood units were transfused to the patients. One segment from each unit of blood was obtained and processed as follows. The blood storage buffer from the collection and the processing typically contains proprietary additive solutions AS1, AS3, or AS7, containing a phosphate-based buffer with added adenosine and other components in the US. These storage buffers are compatible with most typical antibody testing methods. Based on the general calculation, the plasma component containing the antibodies in the blood unit is estimated to be less than 10% (1:10 dilution). One segment of the blood unit contains 0.5 cc leukocyte-reduced red blood cells in AS buffer. Assuming the red-cell concentration in the storage buffer is 50% (250 microliters), and the plasma component in storage buffer is 50% (250 μL). After the segment was obtained, one end of the segment was cut with a scissors and the open end was emptied in a 1.5 mL microfuge tube. Then the other end was cut with a scissors and the blood in the tube was released to the tube entirely. An aliquot of 0.5 mL normal saline was used to wash the remaining blood from the open segment, resulting in approximately 1.0 mL red cells with the buffer (the final dilution is estimated to be 1:40 of the original plasma). The microfuge tube with blood was centrifuged at 10,000× *g* for 1 min, and the supernatant from the tube (50 μL) was directly used for direct ELISA assays.

The plasma samples from the patients with Crohn’s disease (CD) and Sjogren’s syndrome (Sjo) were previously collected through commercial testing services at PZM Diagnostics, and the clinical characteristics of these patients were previously described [18]. The plasma samples were re-tested using the newly established direct ELISA methods using recombinant microbial protein antigens. The plasma samples were diluted 1:50 using normal saline with 5% BSA and the diluted plasma samples were used for direct ELISA assays. The control group was from the physicians’ offices that we have collected over the years with no evidence or history of Crohn’s disease (CD) and Sjogren’s syndrome (Sjo).

## 3. Results

### 3.1. Seroprevalence of Anti-Hsp65 Antibody in Normal Healthy Individuals from the Donor Population of the Red Cross

A total of 288 donor samples were used for the standard direct ELISA assay for detection of the antibodies within the donor samples against the recombinant *Hsp65*. The segment samples were tested using the newly established direct ELISA assays for antibody against *Hsp65* as described in the above sections. The data with all results are summarized including the mean, standard deviation (SD), median, high and low values for each analyte, 95% confidence interval for the mean, cutoff value for each analyte, and the total number and percentage of positive samples in this population based on the cutoff values. The mean for *Hsp65* in 288 Red Cross donors was 0.113 with a standard deviation (SD) of 0.107. The cutoff value for *Hsp65* was 0.327 (based on the mean ± 2× SD). There was a total of eight positive samples for *Hsp65* based on the cutoff value of 0.327 (2.8%).

### 3.2. Hsp65 Antibodies in Crohn’s Disease and Sjogen’s Syndrome

We have collected plasma samples from 109 patients with Crohn’s disease (CD) and from 28 patients with Sjogren’s syndrome (Sjo). A portion of these samples were tested for the same anti-microbial antibodies using indirect ELISA assays with specific capturing antibodies and whole bacterial extracts as previously described [14,18]. Currently we re-tested these samples using the newly established direct ELISA assays and specific recombinant *Hsp65* protein antigens. We have collected 156 controls from the clinics and physicians’ offices, and these controls from the clinics and physicians’ offices were used as controls for Crohn’s disease, although these controls were not entirely normal healthy individuals as a variety of chronic conditions, such as rheumatoid arthritis, thyroid diseases, chronic fatigue syndrome, multiple sclerosis, etc., are known for these controls. These controls were different from those obtained from the Red Cross blood donors, and as we have noted these chronic conditions are invariably related to the abnormal interaction between the human host and its colonizing microbes. As shown in Figure 1, there was a significant increase of *Hsp65* in patients with Crohn’s disease (CD) and Sjogren’s syndrome (Sjo) in comparison to the Red Cross blood donors or other controls from the physicians’ offices (*p* < 0.0001). The prevalence of *Hsp65* antibody in Crohn’s disease patients was 67.9% (74 of 109 patients) and 85.7% in Sjogren’s Syndrome patients (24 of 28 patients). There was also a significant increase of *Hsp65* in the control population from the physicians’ offices in comparison to the Red Cross donor population (*p* < 0.01). In a comparison of Crohn’s disease (CD) and Sjogren’s syndrome (Sjo), the results were not significantly different (*p* = 0.286) (Figure 2).

### 3.3. Receiver Operating Characteristic (ROC) Curve

ROC curve analysis was performed for all the samples, including the controls and the Red cross donors as negative for diseases, and all the samples from Crohn’s disease (CD) and Sjogren’s Syndrome (Sjo) patients as positive for diseases using the ROC curve analysis of the R-package [20] (Figure 3) (http://statistics4everyone.blogspot.com/). The area under the curve (AUC) for *Hsp65* is 0.887. The sensitivity of the *Hsp65* test for Crohn’s disease (CD) patients was 90.4%, and the specificity was 74.8% (Figure 3).

## 4. Discussion

The presence of circulating anti-*Hsp65* antibody in patients with Crohn’s disease and Sjogren’s syndrome, but not in the normal population, indicates lasting and persistent abnormal immune response to mycobacterial exposure in patients. The immune response to mycobacterial exposure is chronic and likely genetically determined, since mycobacteria are widely distributed in the environment such as water and soil. These results raise two important issues in clinical practice. The first and foremost important issue is that if a patient has an elevated anti-*Hsp65* antibody in circulation, this patient will have a spectrum of clinical manifestations, i.e., a positive test for *Hsp65* antibody is diagnostic of clinical disease, but not the risk of contracting the disease. This point is important, although there are unanswered questions of disease specificity. In fact, anti-bacterial antibodies such as typhoid antibodies have been used for diagnosis of bacterial infection since George-Fernand Widal first started using them over a century ago (the Widal test) [21]. The basic principle and the methodology remains similar over the last century, and many clinical diagnostic tests for viral and bacterial infections are still in practical use today. Essentially, the invading bacteria/mycobacteria are considered “foreign” and these microbes elicit the human immune response to produce neutralizing/opsonizing antibodies to stop the pathogenic invasion. Activation of T-cell/B-cell/plasma cells in response to invading pathogens is considered “adaptive immunity” in contrast to the “innate immunity” consisting of the physical barriers, phagocytic systems, and complement systems.

We have surveyed the normal population of blood donors from the Red Cross for the presence of anti-*Hsp65* antibody. Blood donors of the Red Cross are normal healthy individuals that have been screened for infectious diseases, and this normal population is different from other control patients we have collected for other studies. Our results showed that under the normal conditions in the general population the anti-*Hsp65* antibody is uncommon (2.8%) and this low prevalence in the normal population is consistent with the classic concept of human immunity that mycobacterial clearance is through the innate immunity, and the adaptive immunity (B- or T-cell activation) is only required to compensate for the deficiency of the innate immunity in the clearance of mycobacterial invasion [22]. Our data also showed that there are significantly elevated anti-*Hsp65* antibodies in the patients with chronic diseases, such as Crohn’s disease (CD) and Sjogren’s syndrome (Sjo) [18,23]. These chronic “autoimmune” diseases are genetically heterogeneous and polygenic, and the pathogenesis of these diseases is more complex. In our study, the similarity of the *Hsp65* antibody seroprevalence rates in Crohn’s disease (CD), Sjogren’s syndrome (Sjo), and other autoimmune diseases raises questions of disease specificity. It is possible that the disease specificity is conferred through genetic heterogeneity and susceptibility to mycobacteria within the human body and in the environment, but not by the mycobacteria themselves. Understanding the “point of entry” or the barrier systems, including the mucosa and skin as parts of the innate immunity system interacting with mycobacteria in the environment, remains the key [24]. It is also a possibility that mycobacteria in the environment undergo significant changes of virulence factors, becoming pathogenic under unknown circumstances [25].

Identification of anti-*Hsp65* antibody in patients with chronic diseases but not in normal healthy controls raises questions of mycobacterial vaccination using the whole bacterial extracts. Historic efforts to develop a *Mycobacterium tuberculosis* vaccine, like many other bacterial vaccines, have used the whole mycobacterial extracts or whole mycobacteria, inactivated or attenuated, and the whole mycobacterial extract elicits the immune responses and the antibody production in a manner similar to those identified in our patient population, and these antibodies are shown to be highly variable in efficacy to the patients [26,27]. Mycobacterial components or specific mycobacterial proteins, however, can be used for vaccination to produce protective, rather than pathogenic antibodies [28]. It seems again that the human body responds to the invading mycobacteria in specific manners, but this specificity of the immune response does not appear conferred through the invading mycobacteria but the host genetic background, i.e., genetic susceptibility determines the host response to the environmental microbes. It should be noted that viral immunity and viral vaccination is entirely different from bacterial/mycobacterial immunity in which non-specific innate immunity plays a major role, whereas in viral immunity a specific adaptive immunity with memory is shown to be the major protective mechanism [22,29].

Our study showed that anti-*Hsp65* antibody is apparently highly sensitive to chronic diseases such as Crohn’s disease (CD), which has previously been associated with MAP infection and Sjogren’s syndrome (Sjo) and may be indicative of a MAP infection in patients with CD and Sjo. The presence of a circulating *Hsp65* antibody can separate those with an abnormal immune response to the environmental mycobacteria from the normal individuals or other clinical mimics. The value of the anti-*Hsp65* antibody as a diagnostic test to a specific disease type appears to be limited due to the fact that many clinical disease types are found to have elevated anti-*Hsp65* antibody in our data. However, the anti-*Hsp65* antibody test is highly sensitive and reliable to identify the individuals with abnormal immune responses to mycobacteria. These individuals with an abnormal immune response (positive test for anti-*Hsp65* antibody) to mycobacteria can manifest a variety of disease processes in different organ systems, dependent likely upon the genetic susceptibility and the phenotypic response of the host. A small number of reports of low incidence of anti-*Hsp65* antibodies in inflammatory bowel disease was difficult to contemplate in the disease mechanism although the data point to more defective host genetic determinants [30,31].

In the new microbiome era it is known that humans are colonized by many diverse microbes and these microbes are commensal and essential in normal human metabolic functions and development [32]. It is also known that there are numerous microbial DNAs within the normal circulation without living microorganisms and these microbial DNAs are not eliciting the T-cell/B-cell/plasma cell response for antibody production based on the current literature [33]. How our body recognizes the microbial DNA and proteins within our circulation in relation to health and disease remains to be a research topic for many years, and understanding these interactions between the host and the environmental factors provides a basis for a better design of therapeutic agents in various “autoimmune diseases” [34].

## Figures and Tables

**Figure 1 microorganisms-08-00047-f001:**
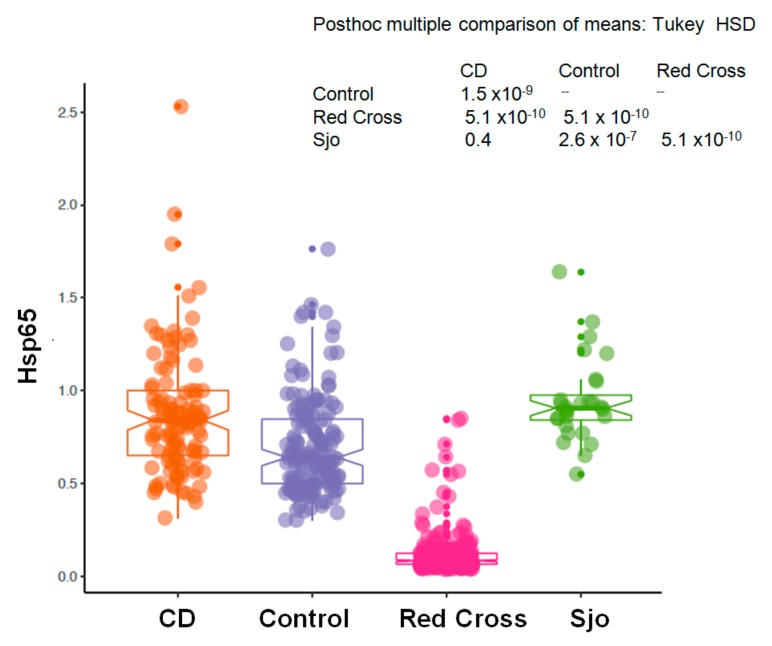
Blood levels of *Hsp65* antibody in a normal population (Red Cross donors, *n* = 288), Crohn’s disease (CD, *n* = 109), Sjogren’s syndrome (Sjo, *n* = 28), and the controls from physician’s office (Control, *n* = 156). Y-axis represents the raw reading unit of OD450.

**Figure 2 microorganisms-08-00047-f002:**
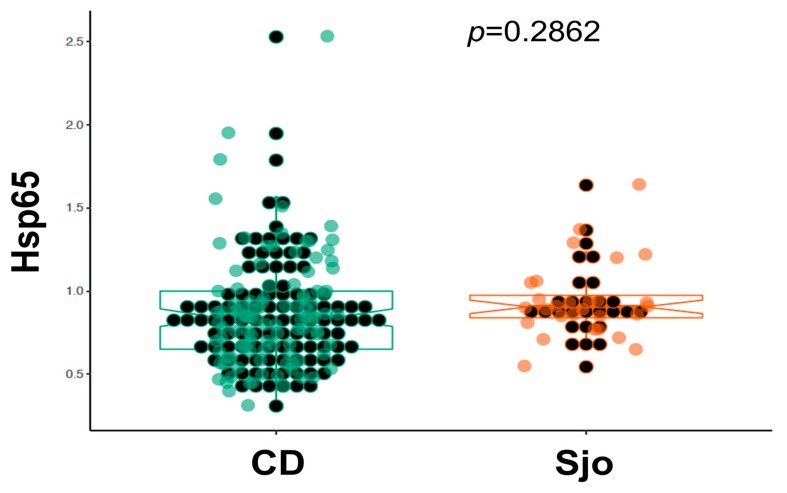
Direct comparison of the blood levels of *Hsp65* antibody in patients with Crohn’s disease (*n* = 109) and Sjogren’s syndrome (*n* = 28).

**Figure 3 microorganisms-08-00047-f003:**
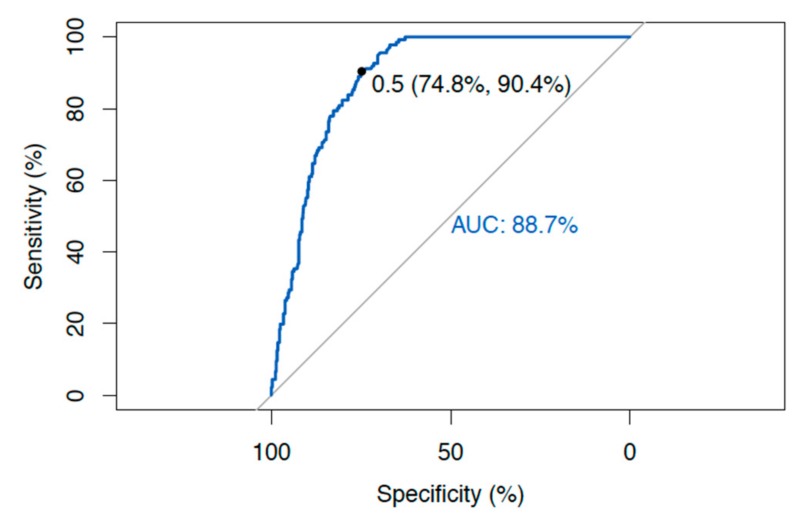
ROC curve analysis of *Hsp65* as a diagnostic test with sensitivity and specificity.

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
