# Peer review of "Serological Testing for Mycobacterial Heat Shock Protein Hsp65 Antibody in Health and Diseases"

_microorganisms, 2019, doi:10.3390/microorganisms8010047_

Round 1
Reviewer 1 Report
Authors present a high-quality experimental manuscript that describes serological testing for Mycobacterial heat shock protein Hsp65 antibody in health and diseases. The manuscript is well-written and a pleasure to read.
Authors surveyed the normal population for seroprevalence of Hsp65 antibody in blood donors of the American Red Cross, and compared the seroprevalence with those of chronic conditions such as Crohn’s disease and Sjogren’s syndrome. 288 donor samples were used for standard direct ELISA assay for detection of the antibodies within the donor samples against the recombinant Hsp65.
Authors observed a significant increase of Hsp65 in patients with Crohn’s disease (CD) and Sjogren’s syndrome (Sjo) in comparison to the Red Cross blood donors or other controls from the physicians’ offices. The results showed that under the normal conditions in the general population the anti-Hsp65 antibody is uncommon (2.8%). The prevalence of Hsp65 antibody in Crohn’s disease patients was 67.9% (74 of 109 patients), and 85.7% in Sjogren’s Syndrome patients (24 of 28 patients). The results are supportive of the view that chronic diseases are infectious diseases in genetically susceptible individuals and the Hsp65 antibodies in circulation represents an abnormal adaptive immune response to antigens of Mycobacterial (environmental) origin.
Based on the results, authors conclude that:
- presence of circulating anti-Hsp65 antibody in patients with Crohn’s disease and Sjogren’s syndrome but not in normal population indicates lasting and persistent abnormal immune response to Mycobacterial exposure in patients
- anti-Hsp65 antibody is rare in the normal population, but frequent in chronic diseases
- presence of circulating Hsp65 antibody reflects an abnormal immune (adaptive) response to Mycobacterial exposure in patients with chronic diseases, thus differentiating the patients with chronic diseases from those clinical mimics.
- the disease specificity is conferred through genetic heterogeneity and susceptibility to Mycobacteria within the human body and in the environment, but not by the Mycobacteria themselves.
- Anti-Hsp65 antibody is apparently highly sensitive to chronic diseases such as Crohn’s disease (CD) which has previously been associated with MAP infection and Sjogren’s syndrome (Sjo) and may be indicative of a MAP infection in patients with CD and Sjo
Other comments:
- the word ’trillions’ (lane 246) is suggested to be rephrased
- please improve resolution of Figures (they are a bit blurry)
Overall, the manuscript is very well suited to be published as an original article in Microorganisms.
In addition, authors are kindly suggested to cite the following article that reviews novel therapeutic targets in autoimmune diseases -
https://doi.org/10.1007/s12668-016-0233-x
Author Response
We appreciated very much for the reviewer's comment, and we made changes as suggested by the reviewer, as highlighted with tracking change in Word. We added 5 more references in respective sections, as recommended by the second reviewer.
The original manuscript with figures at the end in Word was in high resolution. When converted to the PDF files at the editorial office, the PDF file was not optimal with missing figure legends, shifted labeling on the figure 1 on X-axis. That leads to misrepresentation of the data, causing confusion for the reviewer #2.
I have attached the original manuscript and original figures at the end. If needed, I have all the figures in Powerpoint, but I am not able to upload to the website.
Thank you.
Peilin Zhang, MD., Ph.D.

Reviewer 2 Report
The work on "Serological testing for Mycobacterial heat shock 2
protein Hsp65 antibody in health and diseases" basically describes the results of recombinant antigens obtained commercially to perform ELISA from three different groups of people.
The MS is presented in the form of report and needs to be presented as a research article with enough references.
The methods described are not appropriate and the presentation of results need to be looked into.
No attention is paid for the use of abbreviations.
I made comments in the documents in the document.
The MS needs major modifications in every section starting from title to discussion.

Author Response
We appreciated very much for the careful review of our work. The original manuscript in Word was labeled correctly with all the elements. The conversion to PDF files resulted in missing figure legends, shifted labeling on X-axis in Figure 1, and low resolution of the figures.
We added 5 more references in the respective sections of the text as suggested by the reviewer, and we made changes in the text as highlighted by tracking change in Word with some added details in the text. We also corrected errors of abbreviations in the original manuscript.
I hope the reviewer will find the revised manuscript suitable for publication in the journal.
Thank you
Peilin Zhang, MD., Ph.D.
Round 2
Reviewer 2 Report
The pdf file is attached with the comments, the article needs serious look into it.
